# Recent Advances in Smart Hydrogels Prepared by Ionizing Radiation Technology for Biomedical Applications

**DOI:** 10.3390/polym14204377

**Published:** 2022-10-17

**Authors:** Jinyu Yang, Lu Rao, Yayang Wang, Yuan Zhao, Dongliang Liu, Zhijun Wang, Lili Fu, Yifan Wang, Xiaojie Yang, Yuesheng Li, Yi Liu

**Affiliations:** 1Key Laboratory of Coal Conversion and New Carbon Materials of Hubei Province, School of Chemistry and Chemical Engineering, Wuhan University of Science and Technology, Wuhan 430081, China; 2Non-Power Nuclear Technology Collaborative Innovation Center & Hubei Key Laboratory of Radiation Chemistry and Functional Materials, Hubei University of Science and Technology, Xianning 437100, China; 3Hubei Industrial Technology Research Institute of Intelligent Health, Hubei University of Science and Technology, Xianning 437100, China; 4College of Chemistry and Chemical Engineering, Tiangong University, Tianjin 300387, China

**Keywords:** ionizing radiation technology, smart hydrogels, biomedical, cross-linking, biological materials

## Abstract

Materials with excellent biocompatibility and targeting can be widely used in the biomedical field. Hydrogels are an excellent biomedical material, which are similar to living tissue and cannot affect the metabolic process of living organisms. Moreover, the three-dimensional network structure of hydrogel is conducive to the storage and slow release of drugs. Compared to the traditional hydrogel preparation technologies, ionizing radiation technology has high efficiency, is green, and has environmental protection. This technology can easily adjust mechanical properties, swelling, and so on. This review provides a classification of hydrogels and different preparation methods and highlights the advantages of ionizing radiation technology in smart hydrogels used for biomedical applications.

## 1. Introduction

In the 1970s, bioactive glasses with effective tissue binding properties were discovered [1]. With the passage of time, biomedical materials have been gradually updated. Nowadays, high-end materials, such as tissue scaffolds, tissue regeneration, and corneal contact lenses, appear in the biomedical field [2,3,4,5].

Biomedical materials used for human contact need to have the following characteristics: (1) they are biocompatible and (2) they have favorable physical and chemical properties, such as mechanical properties, aging resistance, plasticity, and interfacial stability [6,7,8]. Common biomedical materials are mainly metals, ceramics, and hydrogels. Metals, with high mechanical strength and fatigue resistance, are mainly used to repair hard tissues, such as teeth and bones [9]. However, they are susceptible to corrosion and oxidation. Ceramics can avoid the above disadvantages. In addition, they have favorable high-temperature resistance and osteoconductivity [10]. However, their high brittleness and low toughness severely limit the range of applications. Hydrogels can overcome the shortcomings of the above materials and have been widely studied in the biomedical field.

Hydrogels are 3D networks of polymers formed by physical or chemical cross-linking [11]. Due to their advantages of porosity, stimulus responsiveness, and biodegradability, they can be widely used in agriculture [12], food packaging [13], and optoelectronic materials [14], and especially so in the field of biomedicine [15,16,17]. They are similar to extracellular matrices and provide a favorable environment for tissue regeneration and wound healing [18,19]. There are many methods employed to prepare hydrogels, which can be divided into physical, chemical, and ionizing radiation cross-linking. The hydrogels prepared by physical cross-linking have poor mechanical properties [20]. Chemical cross-linking requires the addition of initiators and cross-linking agents. Moreover, its polymerization efficiency is low and pollutants are easily produced. Radiation cross-linking has the advantages of simple operation, room temperature reaction, and high efficiency [21].

Many researchers have reviewed and prospected the applications of hydrogels in the biomedical field. For example, Chen et al. summarized the application of hydrogels in cell culture, medical surgery, tissue engineering, and biosensing [22]. Liu et al. reviewed the progress of 3D printing hydrogel technology [23]. Taaca et al. studied the preparation methods of hydrogels and emphasized the advantages of a plasma-based preparation of hydrogels [24]. This paper introduces the classification of hydrogels and other preparation methods. The benefits of ionizing radiation technology in the preparation of hydrogels are emphasized. Furthermore, we summarize the recent advances in the fabrication of smart hydrogels for biomedical applications. After searching, no research reporting these contents were found.

## 2. Classification of Smart Hydrogels

Smart hydrogels can exchange energy with the external environment to achieve accurate drug release [25]. They have different degrees of response to external stimuli, so they have been widely studied in the biomedical field. Figure 1 shows the classification of smart hydrogels according to stimulus sources.

### 2.1. Temperature-Sensitive Hydrogels

The swelling degree of thermosensitive hydrogels change with temperature. They contract or expand suddenly around a specific temperature. This temperature point is called volume phase transition temperature (VPTT) [26]. Temperature-sensitive hydrogels are macromolecular chains composed of hydrophobic groups (alkyl groups) and hydrophilic groups (carboxyl, hydroxyl, and amide groups). The ratio of hydrophobic to hydrophilic groups can change the volume phase transition temperature to approach the body temperature. The number of hydrophobic and hydrogen bonds affects the structure of hydrogels, causing them to expand and contract when the temperature changes [27]. Therefore, they can be divided into heat-expandable hydrogels and heat-shrinkable hydrogels.

Around the VPTT, the swelling rate of thermo-swelling hydrogels increases abruptly [28]. Hydrogels formed by polyethylene glycol, methacrylic acid, or acrylic acid all have thermal-swelling properties [29,30,31]. However, thermal-shrinking hydrogels can shrink rapidly at high temperatures, causing the liquid to leave the network structure [32]. N-isopropyl acrylamide (NIPAAm) is a representative monomer for thermal-contracted hydrogels. López-Barriguete et al. prepared hydrogels with different temperature responses using γ radiation at 50 kGy [33]. They selected five feasible systems and presented their low critical solution temperature (LCST), as shown here in Table 1. The LCST value of this hydrogel formed by N-isopropyl acrylamide and dimethyl acrylamide (NIPAAm-co-DMAAm) was 39.8 °C. The LCST of the hydrogels was closest to human body temperature, which laid a suitable foundation for the use of synthesis of temperature-sensitive biosensors in the future.

Li et al. prepared temperature-sensitive color-changing hydrogels by electron beam pre-radiation and radiation cross-linking [34]. By changing the radiation dose and prepolymer composition ratio, the LCST was altered. The LCSTs of NIPAAm/HHPC, NIPAAm/HHPC/Fe_2_O_3_, and NIPAAm/HHPC/GO were 39.5 °C, 37.8 °C, and 41.8 °C, respectively. Color changes could occur in all three hydrogels, both around the LCST (Figure 2). They were expected to replace thermometers.

### 2.2. pH-Responsive Hydrogels

Dissociation of amino, carboxyl, and sulfonic acid groups by adjusting pH can affect the swelling properties of pH-responsive hydrogels [35]. These hydrogels can be classified as the polyacid type and polybasic type. Polyacid hydrogels are formed from cross-linked carboxyl monomers (acrylic acid, methacrylic acid, etc.). When the pH of the dispersion medium is higher than the ionization constant (pK), the carboxyl group loses protons and dissociates to achieve swelling. When the pH of the dispersion medium is lower than the pK of the polybasic hydrogel, the primary group on the side chain of its molecular chain accepts protons. Then, the osmotic pressure inside the hydrogel rises, resulting in swelling [36].

Bustamante-Torres et al. synthesized acrylic and gelatin copolymer hydrogels (AgAR-co-AAC) by gamma irradiation [37]. By measurement, the critical pH points of these hydrogels are about 5.4. The pH was close to that of the forehead and cheek, showing effective biocompatibility [38,39]. When the radiation dose was 20 kGy, AAc was 20%, and the pH value exceeded 5.4. Electrostatic repulsion of polyacrylic acid (PAA) occurred, resulting in rapid swelling of the hydrogel with a water absorption rate of 6000 times (Figure 3a). By simulating physiological conditions, the drug release performance was studied (Figure 3b). Hydrogels with low radiation doses had a weaker cross-linking degree and were easier to promote drug release. In the solution with low ionic force, the small expansion of polymer was not conducive to the release of drugs. The antibacterial experiments on Escherichia coli (*E. coli*) and Staphylococcus aureus (*S. aureus*) demonstrated that the hydrogel-adsorbed ciprofloxacin or silver nanoparticles had a better bacteriostatic effect than blank samples. AgAR-co-AAc hydrogels had a favorable effect against bacteria due to the adsorption of drugs and silver nanoparticles on the hydrogel, which was expected to be a medical dressing for wound healing and skin burns.

### 2.3. Chemical-Responsive Hydrogels

Chemical-responsive hydrogels modulate swelling by interacting with chemicals (glucose, enzymes, antigens, etc.) [40,41,42]. Glucose-responsive hydrogels are particularly effective in treating diabetes. People with diabetes must rely on insulin injections to retain normal blood sugar levels. If too much or too little of a drug is used, the treatment does not work optimally. Glucose-sensitive hydrogels make safe, effective, and lasting drug delivery possible by releasing specific amounts of drugs based on blood glucose levels [43].

Peng et al. prepared cellulose/4-vinylphenyl boric acid (VPBA) hydrogel films by electron beam irradiation [44]. The addition of phenylboronic acid made the hydrogel film dually responsive. Experimental investigation showed that the increase in the number of phenyl borate anions would cause the release of insulin. The films had favorable biocompatibility and could be widely used in sugar-sensitive separation systems.

### 2.4. Light-Responsive Hydrogels

Light-responsive hydrogels are prepared by cross-linking monomers containing photosensitive groups. When they are illuminated by light (visible or ultraviolet), the dipole moment and geometry of the photoactive groups change. This situation results in changes in the 3D network structure, enabling the storage and release of drugs [45].

Cao et al. prepared UV-responsive supramolecular hydrogels by γ-irradiation, which could be used as a drug delivery system for naproxen [46]. The principle of the drug release is that hydrogel undergoes gel-sol transformation under light irradiation. From the quantitative experiments, it was apparent from observing Figure 4 that the color of the hydrogel darkened with the increase in exposure time. It was confirmed by liquid chromatography-mass spectrometry that the ester hydrolysis of o-nitrophenyl was the main driving force for its light response.

### 2.5. Magnetic Field-Responsive Hydrogels

Magnetic field-responsive hydrogels are constituted by doping hydrogels with magnetic materials. Magnetic materials mainly include oxides of iron, cobalt, and nickel. Among them, iron oxide is the principal one. Magnetic particles absorb electromagnetic waves and generate magnetic heat [47]. Magnetic heat promotes circulation, relieves pain and swelling, and can be prescribed for chronic diseases (periodontitis, cervical myelopathy, lumbar spondylosis, scapulohumeral periarthritis, etc.). Without damaging normal cells, it can also inhibit or destroy tumor cells and improve the therapeutic effect [48,49,50]. In addition, nano-magnetic materials can increase the toughness and stability of hydrogels, which makes them commonly used in biomedical fields [51].

Deuflhard et al. embedded iron oxide nanoparticles in gelatin 3D networks by electron beam radiation [52]. The magnetic field-responsive hydrogels were linked to radiation dose, gel concentration, and so on. These causes might affect the degree of equilibrium magnetism. Placing the hydrogel in a magnetic field produces a degree of bending, as showed in Figure 5. The hydrogel shown in Figure 5a was not twisted in a non-magnetic area but bent strongly to the right in the magnetic field (Figure 5b). Therefore, it could be a novel non-contact soft actuation material in the living body by adjusting different influencing causes.

### 2.6. Electric Field-Responsive Hydrogels

Electric fields can be better regulated and applied than other stimulation sources. Electric field-responsive hydrogels are mainly composed of polyelectrolyte materials (polythiophene, polypyrrole, and polyaniline), which can convert electrical and mechanical energy under electric field stimulation [53,54,55]. The macroporous structure of the hydrogel is more likely to cause volume collapse than the microporous structure in the electric field [56]. Furthermore, electroactive efficiency of 3D polymer networks becomes larger with increasing charge density, which can accelerate drug delivery [57].

Chang et al. prepared transparent polyvinyl alcohol (PVA)/polyethylene glycol diacrylate (PEGDA)/agar/sulfuric hydrogels with high conductivity and self-healing by irradiation [58]. The ionic conductive hydrogel was more flexible than traditional electronic devices. The hydrogel displayed a unique electrical current when undergoing mechanical motion, which could serve as sensors in vitro.

## 3. Preparation Method of Smart Hydrogels

Smart hydrogels are mainly composed of monomers or nanoparticles with unique properties. The preparation methods can be divided into physical cross-linking, chemical cross-linking, and radiation cross-linking (Table 2).

### 3.1. Physical Cross-Linking

Hydrogels prepared by physical cross-linking are formed through non-covalent interactions, such as hydrogen bonding and intermolecular and hydrophobic forces [69,70,71]. Due to reversibility and fluidity, they become excellent injectable materials.

Injectable hydrogels are extensively used in drug delivery and tissue engineering due to their degradability and biocompatibility. Li et al. prepared injectable hydrogels using pH-responsive octapeptide (FOE) as loading material [59]. The new types of hydrogels had the advantages of favorable physical and chemical properties, which could concentrate the drug on the tumor site and reduce the side effects in the body. The mechanical properties of hydrogels prepared by extracellular derivatives, such as collagen, fibrin, and hyaluronic acid, were poor, which affected their application in tissue engineering. Zhao et al. used calcium ion cross-linked PVA to form a robust framework [60]. Then, a double-network hydrogel was obtained by adding bioactive glass microspheres and poly(ethylene glycol). After mineralization for 14 days, the compressive strength, modulus, and fracture energy of the hydrogel can reach 57 MPa, 2 MPa, and 65 kJm^−2^, respectively. In addition, the effect of the injected hydrogel in the treatment of bone defects was better than that of implanting a large volume of hydrogel.

### 3.2. Chemical Cross-Linking

Compared to physical cross-linking, hydrogels prepared by chemical cross-linking has high stability and can flexibly change pore size. Chemical cross-linking is a process in which monomers generate free radicals for cross-linking polymerization [72]. According to different methods of generating primary free radicals, chemical cross-linking can be divided into initiator cross-linking, photo-initiated cross-linking, and “click chemistry” cross-linking.

#### 3.2.1. Initiator Cross-Linking Method

Initiator cross-linking mainly produces free radicals that initiate, grow, terminate, and transfer chains to form hydrogels [73]. The initiators that promote the production of free radicals mainly include azo, organic peroxides, inorganic peroxides, and redox initiator systems. Among them, persulfate in inorganic peroxides is the most common. Kurdtabar et al. used graphene oxide (GO) and sodium carboxymethyl cellulose (CMC) as raw materials [61]. At high temperatures, sulfate anion radicals triggered the formation of a GO/CMC-g-PAA copolymer from AA monomers. The magnetic iron oxide nanoparticles (MIONs) were made by adding Fe^2+^/Fe^3+^ to the GO/CMC-g-PAA solution. MIONs chelated multiple polymer chains to form multi-responsive hydrogels for the precise release of anticancer drugs (Figure 6).

#### 3.2.2. Photo-Initiated Cross-Linking Method

Photo-initiated cross-linking mainly promotes monomer cross-linking by generating free radicals from photo-initiators [74,75]. Under intense light irradiation, the monomers can rapidly polymerize [76]. Conventional photo-initiators have specific toxicity, which affects their application in biomedical fields. Graphene quantum dots (GQDs), which are non-toxic and have a broad spectral absorption range, are ideal photo-initiators. Kim et al. proposed the preparation of polyamide hydrogels using GQDs as a photo-initiator [62]. Their elastic modulus was 50 times higher and the swelling ratio was similar to that of conventional photo-initiated hydrogels. Most importantly, they had a light transmittance of more than 90% and were expected to be the primary materials for contact lenses.

#### 3.2.3. “Click Chemistry” Cross-Linking Method

Compared to the above methods, “click chemistry” cross-linking can protect unique functional groups [77]. As the amino group of chitosan reacts quickly, the pH-responsive function of chitosan hydrogel is lost. Therefore, Ding et al. endowed chitosan hydrogel with a UV cross-linking ability and pH responsiveness through “thiol-ene” click chemistry technology [78]. It gelatinized within 30 s under UV light of 4 mW/cm^2^. Subsequently, Wiwatsamphan et al. prepared dual-network pH-/heat-responsive chitosan/poly (N-isopropylacrylamide) hydrogels (CN-IPNs) by thiol-ene clicking [63]. CN-IPNs had suitable mechanical properties and durability, which could be used in artificial muscles.

### 3.3. Radiation Cross-Linking Method

Ionizing radiation technology is now very active and widely used in agriculture, medicine, and the environment [79,80,81]. The irradiation process is carried out at a normal temperature and the operation is simple and easy to control. There is no need to add any initiator and the product is highly pure. At the same time, irradiation has a bactericidal function, making it more suitable for research in the biomedical field [82,83]. Irradiation cross-linking can be utilized to form polymers from free radicals initiated by electron beams, γ-rays, and X-rays [84]. Electron beam radiation uses the ionization and excitation effects of electrons and matter to prepare polymers [85]. Electron accelerators have the advantages of high radiation utilization, high dose rate, and no radiation after power failure. They are an excellent production device and are favored by researchers. Compared to the electron beam, γ-rays have a high penetration into matter and can be irradiated for a long time. Radiation preparation methods of hydrogels can be divided into radiation cross-linking, radiation polymerization, and radiation graft copolymerization.

#### 3.3.1. Radiation Cross-Linking

The radiation cross-linking method is used to induce the cross-linking reaction between polymer chains. The hydrogels prepared by the radiation cross-linking method have favorable swelling, which is important in the biomedical field. Lugo-Medina et al. synthesized two NIPAAm hydrogels by chemical cross-linking and electron beam irradiation [64]. The swelling and shrinkage tests of two types of samples in water and different solutions observed in the swelling and shrinkage of gels prepared by the irradiation method were much higher than those prepared by the chemical cross-linking method. Furthermore, chemically cross-linked hydrogels behaved as polyelectrolytes. However, the hydrogels irradiated by electron beam showed amphiphilicity in different salt solutions. At the same time, they were sensitive to temperature changes. The preparation of hydrogels by electron beam radiation is a suitable method for drug delivery.

#### 3.3.2. Radiation Polymerization

Radiation polymerization is a cross-linking method that generates free radicals from monomers [86]. Amin et al. prepared temperature and pH-responsive hydrogels by electron beam irradiation using bacterial cellulose (BC) and acrylic acid (AA) as monomers [65]. The electron beam induced water molecules to produce free radicals. Hydrogen atoms and hydroxyl radicals initiated AA to form a PAA copolymer. Hydrogen atoms promoted BC to generate active sites and cross-link with PAAc to form a network structure.

Hu et al. prepared MXene-based nanocomposite hydrogels using 2-(dimethylamino) ethyl methacrylate (DMAEMA) and ultra-low content Ti_3_C_2_T_x_ MXene nanosheets as the main materials [66]. DMAEMA was cross-linked into polymers by gamma radiation. At the same time, Ti_3_C_2_T_x_ interacted with polymer chains through hydrogen bonds and covalent bonds. The electrical conductivity of the composite hydrogels was considered to be 1.6 mScm^−1^. At the same time, they had specific self-healing abilities and were expected to be used as excellent materials for electronic skin.

#### 3.3.3. Radiation Grafting

Radiation grafting is one way to make the polymer produce radicals for graft copolymerization [87]. Hydrogel preparation and membrane alteration are the two main applications of this technique. Pitarresi et al. used γ radiation to combine aqueous PNG with N, N-methylene bisacrylamide (BIS) to form a hydrogel network [67]. Gel particles containing the drugs were released in a solution of pH = 1 or 7.4, which could be used in topical treatments. Bardajee et al. synthesized smart hydrogels by graft copolymerizing acrylic acid (AA) with the herbaceous skeleton [88]. A suitable water absorption rate could be obtained by changing the variables. In addition, the hydrogels were responsive to medium pH, salt solution, and mixed solvent. Crotonic acid (CrA) could hardly react with polymers. Therefore, Ajji et al. grafted and polymerized polyvinylpyrrolidone (PVP) and crotonate by γ-radiation grafting [68]. The drug release experiment showed that ketoprofen could be released in the neutral medium for a long time. The hydrogels might be applicable to the targeted release of drugs in the intestine.

## 4. Application of Smart Hydrogels

Hydrogels are a class of hydrophilic 3D network structures with effective biocompatibility and non-toxicity and have become candidate materials in the biomedical field (Figure 7). For example, smart hydrogels have been widely studied in dressings, drug carriers, regeneration medicine, and medical devices (Table 3).

### 4.1. Hydrogel Dressings

Bacterial infections have caused severe harm to human life and health. Overuse of antibiotics has resulted in the emergence of drug-resistant strains. The development of green antibacterial materials has grown. Researchers have developed a series of new, efficient, and non-toxic antibacterial dressings, which play an irreplaceable role in antibacterial and anti-infection effects. Mozalewska et al. prepared a wound dressing based on PVP and agar by irradiation [89]. Microbiological experiments showed that dressing could inhibit the increase in gram-positive bacteria. Gao et al. made a series of PVA hydrogels loaded with silver sulfadiazine (AgSD) by electron beam irradiation [90]. The AgSD/PVA hydrogels showed excellent antibacterial activity against *E. coli* and *S. aureus*. They might be an ideal dressing for antibacterial wounds. Li et al. prepared PVA/CMCS/TiO_2_ ternary nanocomposite hydrogels by freeze-thaw cycles and electron beam radiation [91]. The composite hydrogels showed excellent antibacterial activity against *E. coli* and *S. aureus* and no cytotoxicity. The photosensitive antibacterial hydrogels made by our group had great application potential in medical dressings (Figure 8). Collagen/PVP/PAA/polyethylene oxide (PEO) hydrogels were obtained by electron beam irradiation in an inert atmosphere. They had high water absorption, elasticity, and stability, and were used in soft tissue engineering. Rheological experiments confirmed that the collagen/PVP/PAA/PEO hydrogels had rheological behaviors similar to most soft tissues. The hydrogels maintained adequate stability and physical morphology in the pH environment of intact skin and broken wounds [104].

As early as ancient Greece and Rome, honey was reported to be used as an antiseptic to treat wounds. It mainly achieved therapeutic effects by exploiting its high osmotic activity [105]. Nho et al. prepared honey hydrogels with carboxymethylcellulose (CMC) and honey as raw materials [92]. They showed excellent antibacterial activity against *S. aureus* and *E. coli*. The animal experiments showed that the hydrogel containing honey could accelerate wound healing. Jeong et al. designed a PAA hydrogel loaded with different metronidazole (MD) contents [93]. When the absorbed dose was 25 kGy, the MD/PAA hydrogel exhibited certain features. For instance, sufficient gel content and strength could be used as a dressing. About 80% metronidazole in PAA hydrogels could be released within 120 min. MD/PAA hydrogels were non-cytotoxic and had excellent antibacterial activity against *S. aureus* and *E. coli*. Singh et al. prepared hydrogels for skin burn dressings using gamma radiation [106]. Nanosilver hydrogels were effective in controlling microbial infection and promoting the healing of burn wounds. In addition, γ-irradiation can be synergistic with other techniques to prepare multifunctional hydrogels. Liu et al. synergistically prepared gelatin and γ-polyglutamic acid (γ-PGA) hydrogels using hot pressing and γ-irradiation methods [94]. They exhibited effective biocompatibility, biodegradability, and mechanical strength. In addition to electron accelerators, ^60^Co radioactive sources are also a suitable means of radiation. Lim et al. prepared hydrogels for the treatment of dermatitis [107]. It had been proved by animal experiments that the hydrogel dressing was effective for dermatitis. Alcântara et al. prepared a PVP/PVA hydrogel by gamma irradiation with a ^60^Co source [108]. These hydrogel dressings had potential application in wounds infected with gram-positive and gram-negative bacteria. Overall, hydrogel dressings are more readily available than other materials. However, most researchers have only studied the antibacterial effects of *S. aureus* and *E. coli*. They have not learned the whole antibacterial spectrum, which is an area that needs to be strengthened in the future.

### 4.2. Drug Carriers

Drug carriers deliver the drug to the designated area at a certain time and speed, which can reduce the adverse effects of the drug on the body. Drug delivery systems have been gradually developed, mainly using polymeric materials, such as hydrogels, as drug carriers. El-Rehim et al. prepared low-viscosity PVP/PAAc nanogels by gamma irradiation [95]. They did not cause blurred vision or blindness, or other side effects. In vitro release studies showed that PVP/PAAc nanogels had a longer sustained release time for maurocasone. Therefore, they could improve the availability of drugs. Ishak et al. prepared pH-sensitive nanocrystalline fiber/gelatin hydrogels [109]. Using riboflavin as a drug model, the release results showed that more than 70% of the riboflavin in the gelatin/4CNC hydrogel was released within 12 h. It was non-cytotoxic and had potential application potential in drug delivery systems.

Drug carriers can target different parts of the body. Among them, drug release from the gastrointestinal tract is the most frequent. Bhuyan et al. synthesized pectin/N, dimethyl acrylamide (DMAA) hydrogels by γ-radiation [96]. 5-fluorouracil was used as the simulated drug. Under the pH value of gastric juice and intestinal juice, the drug release after four hours could reach more than 90%. Raafat et al. prepared pH-sensitive hydrogels based on gelatin and acrylic acid (AAc), which were polymerized and cross-linked using gamma radiation [110]. Ketoprofen was used as a model drug to study the release of hydrogel in vitro. The swelling kinetic studies showed that the hydrogels had Fick diffusion in the stomach (pH = 1), and non-Fick distribution in the intestine (pH = 7). The hydrogels were recommended as site-specific drug carriers. In addition, poly(ethylene oxide) (PEO) network grafted AAc hydrogels were prepared by a two-step method. These pH-sensitive hydrogels were investigated as drug carriers to protect insulin from the acidic environment of the stomach [111].

Chemotherapy is the primary means of cancer treatment. While killing cancer cells, normal cells also suffer significant damage. If chemotherapy drugs could be targeted to specific cancer cells, they would greatly contribute to human health. Nisar et al. prepared glutamate-grafted chitosan (CHG-GA) hydrogel microspheres used to carry the anticancer drug doxorubicin by γ-radiation grafting method [97]. The pH response value of the hydrogel was 5.8, which was close to the physiological pH value of tumor tissue. The drug release rate was the highest (81.33%). The CHG-GA hydrogel microspheres had favorable biocompatibility and the cell viability was approximately 95%. The CHG-GA hydrogel microspheres loaded with doxorubicin had an anticancer effect on MCF-7 cells. They had a broad application prospect in the controlled delivery of anticancer drugs for local cancer treatment. Glass et al. used electron beams to synthesize transparent hydrogels for drug-loading applications [112]. These hydrogels had higher mechanical properties, optical transmittance, and cross-linking density than conventional hydrogels prepared by UV light. Using methylene blue as a drug model, the content of methylene blue in electron-beam-polymerized hydrogels was twice that of UV-polymerized hydrogels. Park et al. developed β-glucan hydrogels that could be used to treat periodontal disease [113]. According to the cytotoxicity and antibacterial activity test, the hydrogels had no cytotoxicity and effective antibacterial activity. They could be used in continuous drug release and the drug release rate could reach 80% in about 90 min. Hydrogel carriers provide a new method for clinical treatment. However, the controlled release data for hydrogels are not accurate. Some drugs are released too early to take effect.

### 4.3. Regenerative Medicine

The use of engineering methods to repair damaged organs and tissues has opened new avenues for regenerative medicine. Wach et al. prepared CMCS hydrogels by radiation graft cross-linking, which could be used as potential intracavity-filling nerve-regeneration channels [98]. Cytotoxicity tests and in vivo tests proved that the hydrogel had effective biocompatibility and antibacterial activity. In addition, carboxymethyl chitosan could be combined with biologically active substances, such as therapeutic drugs or growth factors for peripheral nerve repair. Szafulera et al. prepared a series of glucan derivatives by the ionizing radiation method [99]. The methacrylic acid substitution degree (DS) of dextran methacrylate (Dex-MA) could reach 1.13. Irradiation of methacrylic acid dextran in an aqueous solution was an efficient method to prepare biocompatible hydrogels. These dextran-based hydrogels had broad application prospects in the field of biomedicine, especially in the field of soft tissue regenerative medicine. As one of the issues being explored in regenerative medicine, the implantation of artificial tissues or organs is full of opportunities and challenges.

### 4.4. Medical Devices

Medical devices are one of the core components of medical engineering. In addition to drugs, medical devices are also essential. Medical devices can detect diseases and help the doctor better treat patients.

Contreras-Garcı’a et al. successively grafted dimethyl acrylamide (DMAAm) and NIPAAm onto polypropylene (PP) membranes [100]. (PP-g-DMAAm)-g-NIPAAm graft could improve the hemocompatibility and elution performance of antimicrobial drugs for medical devices. These functionalized PP films had potential as medical devices and drug delivery. Zhu et al. prepared poly(N-isopropyl acrylamide) (PNIPAM)/graphene oxide (GO) hydrogels by γ-ray polymerization [101]. The PNIPAM/GO hydrogels exhibited excellent photo-thermal properties, and their phase transition could be remotely controlled by near-infrared (NIR) laser irradiation. They had broad application prospects in biomedicine, especially in microfluidic devices. Dispenza et al. prepared PVA/PANI hydrogel nanocomposites by electron beam irradiation, which was non-cytotoxic and had potential value for sensing and smart drug delivery applications [102]. Hiroki et al. prepared HPC-based hydrogels with suitable mechanical properties and transparency by electron beam irradiation [114]. They could be degraded by cellulases and used to manufacture contact lenses. With the development of the medical industry, medical equipment needs to keep pace with the times. Yang et al. prepared silver-loaded chitosan and PVA fingerprinting gel films (Ag/CMCS/PVA) by electron beam irradiation [103]. It was found that the fingerprints on the hydrogel prepared by electron beam irradiation were more clearly visible through experimental investigations (Figure 9). Since such hydrogels were non-toxic, they were expected to be used for fingerprint extraction and preservation.

## 5. Conclusions

In summary, the adhesion, degradability, and biocompatibility have promoted smart hydrogels in profound biomedical field research. This article summarized the classification of smart hydrogels and their preparation methods, especially for applying wound dressings, as drug carriers, for regenerative medicine, and for medical devices.

These smart hydrogels act on the human body and their biocompatibility and therapeutic effects have always been the focus of researchers. Compared with other methods, the radiation method is more suitable for the biomedical field due to its favorable sterilization effect. At the same time, ionizing radiation technology is convenient for mass production, which lays a foundation for the industrialization of biomedical hydrogels in the future.

At present, in vivo research of biomedical hydrogels is scarce, which limits its development in the biomedical field. If the above shortcomings can be quickly overcome, bio-intelligent hydrogels will be widely used and contribute to the progress of medicine.

## Figures and Tables

**Figure 1 polymers-14-04377-f001:**
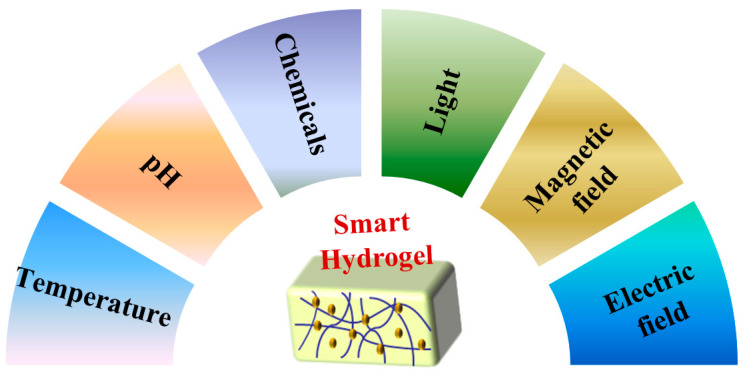
Classification of smart hydrogels.

**Figure 2 polymers-14-04377-f002:**
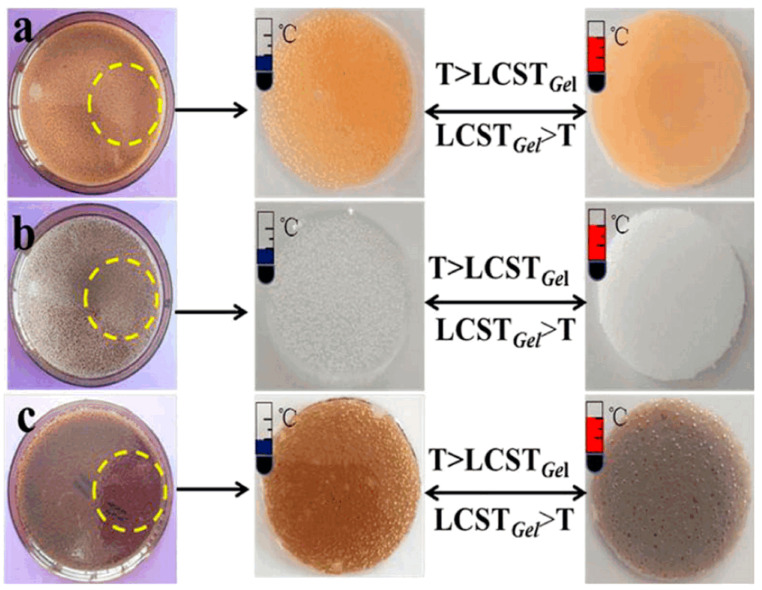
Reversible temperature-sensitive color change process of hydrogels [(**a**) NIPAAm/HHPC/Fe_2_O_3_ (25), (**b**) NIPAAm/HHPC (30), (**c**) NIPAAm/HHPC/Fe_2_O_3_ (35)] [34].

**Figure 3 polymers-14-04377-f003:**
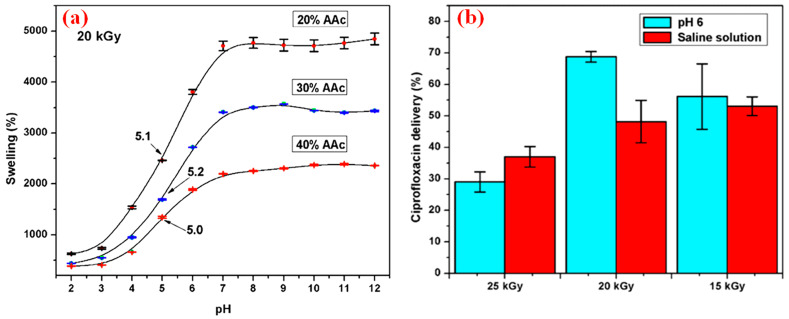
(**a**) Swelling rate of cross-linked hydrogels containing 20% AAc at 20 kGy and at different pH values; (**b**) effects of irradiation dose and solution type on sustained release of ciprofloxacin [37]. Reprinted with permission from Ref. [37].

**Figure 4 polymers-14-04377-f004:**
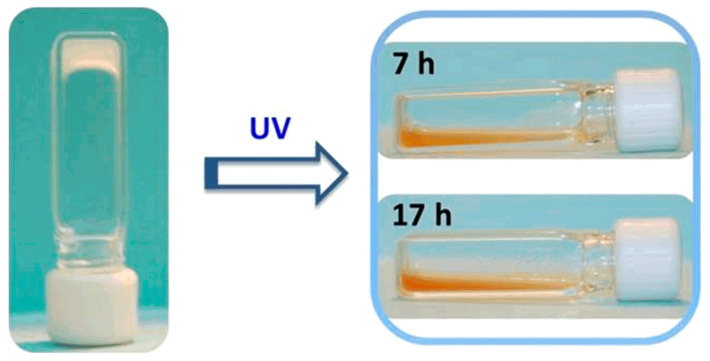
Photographs of hydrogels before UV light exposure and at different light exposure times [46].

**Figure 5 polymers-14-04377-f005:**
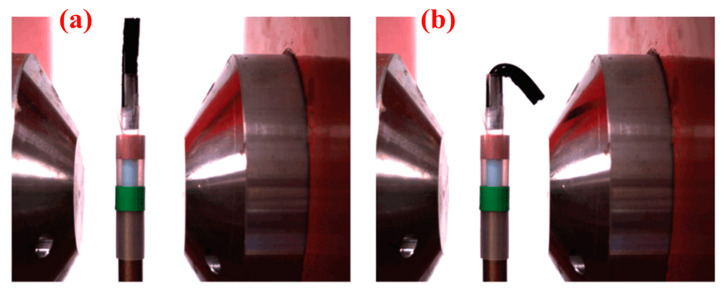
Schematic illustration of reversible bending of an iron gel: (**a**,**b**) show the bending degree of field off and on, respectively [52].

**Figure 6 polymers-14-04377-f006:**
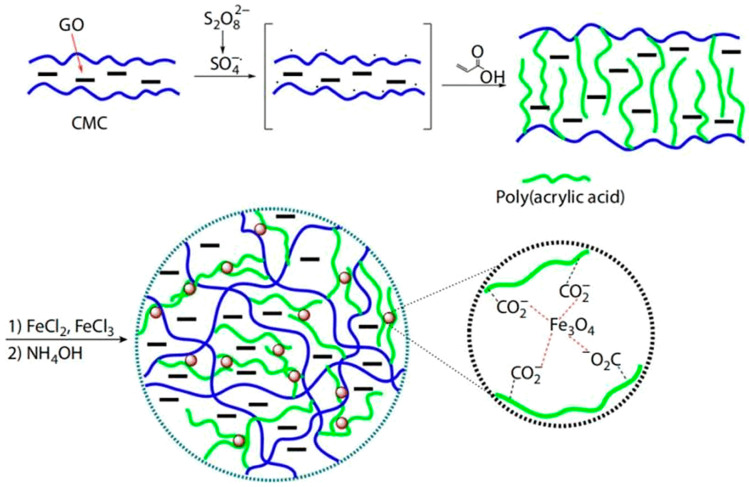
Synthesis roadmap of MION/GO/CMC-g-PAA [61].

**Figure 7 polymers-14-04377-f007:**
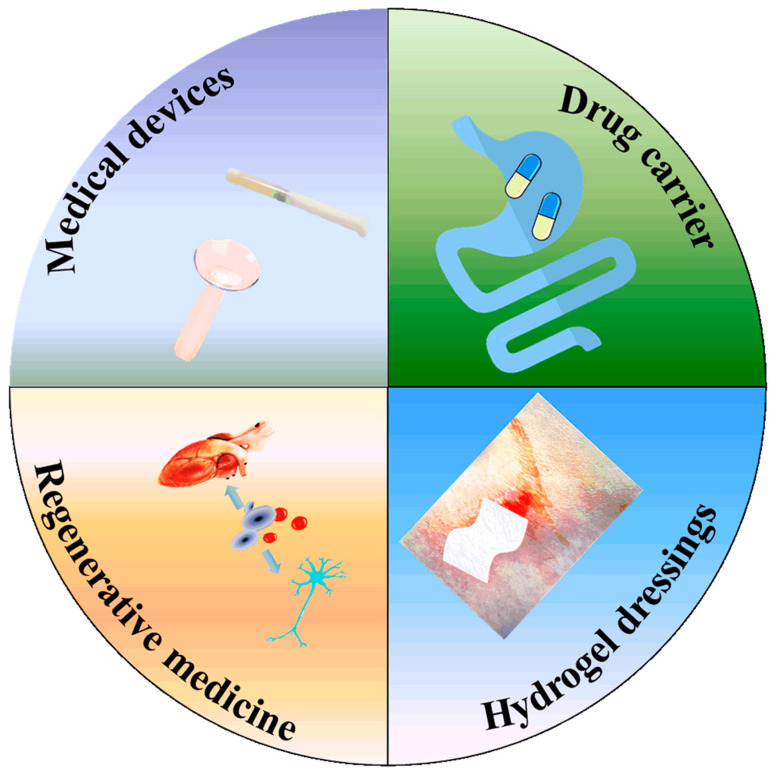
Application of intelligent hydrogel prepared by ionizing radiation technology in regenerative medicine, drug carriers, and wound dressings.

**Figure 8 polymers-14-04377-f008:**
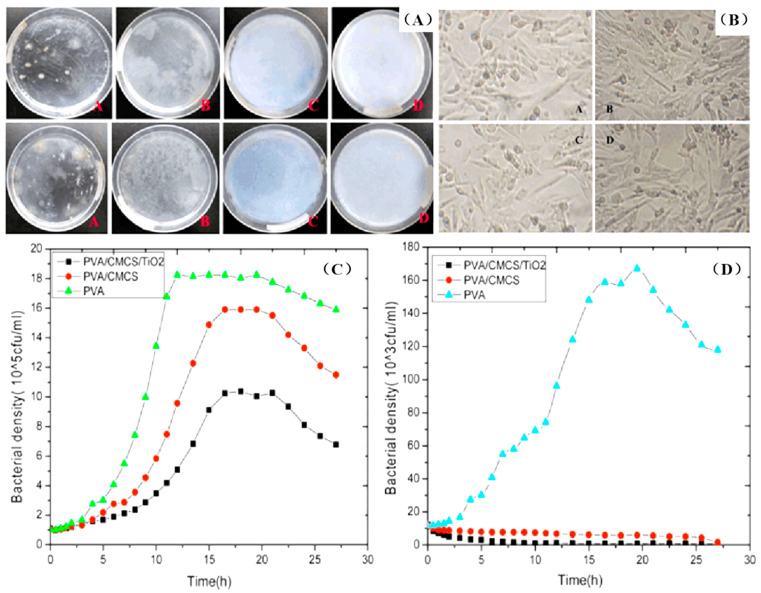
Bacterial activity and cell cytotoxicity of PVA/CMCS/TiO_2_ ternary nanocomposite hydrogels. (**A**) Colony distribution image of different hydrogels on Escherichia coli (top) and Staphylococcus aureus (bottom); (**B**) cytotoxicity of different hydrogels on L929 cells; bacterial density curves of different hydrogels against *E. coli* (**C**) and *S. aureus* (**D**) [91].

**Figure 9 polymers-14-04377-f009:**
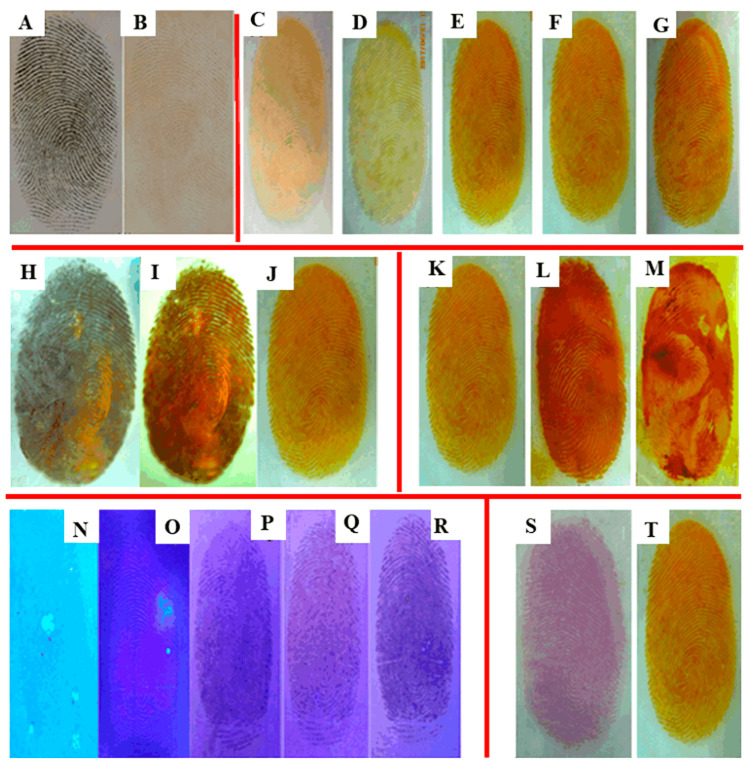
Fingerprint images of PVA/CMCS composite hydrogel film under different conditions. At different excitation wavelength of 254 nm (**A**) and 356 nm (**B**); at different irradiated times under 254 nm UV light: (**C**) 5 min, (**D**) 10 min, (**E**) 20 min, (**F**) 30 min, (**G**) 40 min; at different preparing methods: (**H**) frozen without irradiation, (**I**) refrigeration and irradiated, (**J**) short-term frozen and irradiation; at different coating modes: (**K**) spray, (**L**) smear, (**M**) soak; At different concentration of Ag ion: (**N**–**R**), 0.12 mg/mL, 0.6 mg/mL, 1.2 mg/mL, 1.8 mg/mL, 2.4 mg/mL, respectively; different developing reagents: (**S**) ninhydrin, (**T**) Ag ion [103].

**Table 1 polymers-14-04377-t001:** Experimental codes, LCST results, water absorption, and properties of copolymers obtained at 50 kGy radiation [33].

Sample Poly (A-co-B)	Experimental Codes	A:B Ratio (vol%)	LCST (°C)	Water Absorption (%)	Physical Properties
Poly (NIPAAm-co-AAc)	NAA	50:50	52.2	596	Solid white film
		70:30	43.9	625	Fragile white film
		80:20	48.9	620	Fragile white film
		90:10	45.5	615	Fragile white film
Poly (NIPAAm-co-DMAAm)	ND	50:50	57.0	896	Solid clear film
		70:30	50.5	885	Solid clear film
		80:20	39.8	875	Solid clear film
		90:10	--	885	Solid clear film
Poly (NIPAAm-co-MAAc)	NMA	50:50	56.0	690	Solid white film
		70:30	48.2	695	Solid white film
		80:20	42.7	680	Fragile white film
		90:10	50.9	675	Fragile white film
Poly (NIPAAm-co-HEMA)	NHE	50:50	57.4	542	Solid white film
		70:30	48.0	548	Solid white film
		80:20	49.9	530	Solid white film
		90:10	54.6	528	Fragile white film
Poly (NVCL-co-DMAAm)	VD	50:50	49.6	1002	Solid yellow film
		70:30	47.8	1008	Solid yellow film
		80:20	52.2	995	Flex yellow film
		90:10	--	--	Liquid

**Table 2 polymers-14-04377-t002:** Classification of preparation methods.

Classification	Hydrogel Materials	Preparation Methods	Application	References
**Physical cross-linking**	FOE	Self-assembly method	Drug release	[59]
	PVA/BG/PEG	Ion cross-linking method	Treatment of bone defects	[60]
**Chemical cross-linking**	GO/CMC-g-PAA	Initiator cross-linking method	Drug release	[61]
	AA/N, N’-methylenebisacrylamide	Photo-initiated cross-linking method	Contact lenses	[62]
	CN-IPNs	“Clickchemistry” cross-linking method	Artificial muscle	[63]
**Radiation cross-linking**	NIPAAm	Radiation cross-linking	Drug release	[64]
	AA/BC	Radiation polymerization	Drug release	[65]
	DMAEMA/Ti_3_C_2_T_x_	Radiation polymerization	Electronic skin	[66]
	PNG/BIS	Radiation grafting	Drug release	[67]
	CrA/PVP	Radiation grafting	Drug release	[68]

**Table 3 polymers-14-04377-t003:** Application of smart hydrogels.

Application	Hydrogels	Radiation Resource	References
**Hydrogel dressings**	PVP/ager	Electron beam	[89]
	AgSD/PVA	Electron beam	[90]
	PVA/CMCS/TiO_2_	Electron beam	[91]
	CMC/honey	Gamma rays	[92]
	MD/PVA	Gamma rays	[93]
	Gelatin/γ-PGA	Gamma rays	[94]
**Drug carriers**	PVP/PAAc	Gamma rays	[95]
	Pectin/DMAA	Gamma rays	[96]
	CHG-GA	Gamma rays	[97]
**Regenerative medicine**	CMCS	Electron beam	[98]
	Glucan derivatives	Electron beam	[99]
**Medical devices**	(PP-g-DMAAm)-g-NIPAAm	Gamma rays	[100]
	PNIPAAM/GO	Gamma rays	[101]
	PVA/PANI	Electron beam	[102]
	Ag/CMCS/PVA	Electron beam	[103]

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
