# Peer review of "Recent Advances in Smart Hydrogels Prepared by Ionizing Radiation Technology for Biomedical Applications"

_polymers, 2022, doi:10.3390/polym14204377_

Round 1

Reviewer 1 Report

In the article: “Recent Advances on Smart Hydrogels Prepared by Ionizing Radiation Technology for Biomedical Applications” the authors provide a classification of hydrogels and different preparation methods highlighting the advantages of ionizing radiation technology in smart hydrogels used for biomedical applications. 

Overall, this work is interesting as it is a wide scientific overview, however, we would like to invite the authors to clarify some points:

 1.        It is not clear if this document is an article or a review, thus the journal reports “article” and the authors say “review”. Please, clarify this point;

2.       Among the introduction sections, the authors introduce some general concept. During the description of biomaterials, their potential applications should be thorough, e.g. tissue regeneration, diseases managements. In this field this recent reference should be useful: Vassallo V, Tsianaka A, Alessio N, Grübel J, Cammarota M, Tovar GEM, Southan A, Schiraldi C. Evaluation of novel biomaterials for cartilage regeneration based on gelatin methacryloyl interpenetrated with extractive chondroitin sulfate or unsulfated biotechnological chondroitin. J Biomed Mater Res A. 2022 Jun;110(6):1210-1223. doi: 10.1002/jbm.a.37364. Epub 2022 Jan 28. PMID: 35088923; PMCID: PMC9306773;

3.       In the paragraph “Temperature-sensitive hydrogel”, the authors say that specific hydrogel can replace thermometers. Are they so accurate?

4.       In the field of tissue regeneration, specifically cartilage regeneration, only biocompatibility and cytotoxicity tests are available? Gene and/or protein expression of specific biomarkers have not been in vitro assayed?

5.        In the conclusion section the authors should resume the features of an “intelligent Hydrogel”.

Author Response

Dear reviewer, thank you very much for your valuable comments on this review. Here are the changes I made. Point 1: It is not clear if this document is an article or a review, thus the journal reports “article”  and the authors say “review”. Please, clarify this point; Response 1: This is a review, and we make changes in it. (in red) Point 2: Among the introduction sections, the authors introduce some general concept. During the description of biomaterials, their potential applications should be thorough, e.g. tissue regeneration, diseases managements. In this field this recent reference should be useful: Vassallo V, Tsianaka A, Alessio N, Grübel J, Cammarota M, Tovar GEM, Southan A, Schiraldi C. Evaluation of novel biomaterials for cartilage regeneration based on gelatin methacryloyl interpenetrated with extractive chondroitin sulfate or unsulfated biotechnological chondroitin. J Biomed Mater Res A. 2022 Jun;110(6):1210-1223. doi: 10.1002/jbm.a.37364. Epub 2022 Jan 28. PMID: 35088923; PMCID: PMC9306773;  Response 2: I quite agree with your opinion and have quoted the reference of appeal in the preface of the paper. (in red) Point 3: In the paragraph “Temperature-sensitive hydrogel”, the authors say that specific hydrogel  can replace thermometers. Are they so accurate?  Response 3: Thank you very much for your comments! The statement was so absolute that we changed it to "They are expected to replace thermometers." (in red) Point 4: In the field of tissue regeneration, specifically cartilage regeneration, only biocompatibility  and cytotoxicity tests are available? Gene and/or protein expression of specific biomarkers have not been in vitro assayed?  Response 4: Thank you very much for your opinion, because ionizing radiation method has been studied by a limited number of people. At the same time, the paper focused on the preparation and characterization of hydrogels, and some articles involved cytotoxicity. However, gene and/or protein expression of specific biomarkers was biased towards the biomedical domain and was not found in the cited articles. We regret that we cannot add to this content. Point 5: In the conclusion section the authors should resume the features of an “intelligent  Hydrogel”. Response 5: In the conclusion, we supplement the characteristics of smart hydrogel as you suggested. (in red)

Reviewer 2 Report

The review article entitled "Recent Advances on Smart Hydrogels Prepared by Ionizing Radiation Technology for Biomedical Applications" is of interest to some readers. But the major drawback of this article is there is no figures or tables are presented in the review article. The authors have claimed that they have collected more than 100 papers and summarize the results. But without the image or summery table, it is very hard for the reader to involve in the paper. Hence I suggest the authors to include at least 6-8 figures and 3 tables to attract the readers. 

Author Response

Dear reviewer, thank you very much for your valuable comments on this review. Here are the changes I made. Point 1: The review article entitled "Recent Advances on Smart Hydrogels Prepared by Ionizing  Radiation Technology for Biomedical Applications" is of interest to some readers. But the major drawback of this article is there is no figures or tables are presented in the review article. The authors have claimed that they have collected more than 100 papers and summarize the results. But without the image or summery table, it is very hard for the reader to involve in the paper. Hence I suggest the authors to include at least 6-8 figures and 3 tables to attract the readers. Response 1: Thank you very much for your comments on this article. Nine images and two tables have been added to the text.

Reviewer 3 Report

The review article titled "Recent Advances on Smart Hydrogels Prepared by Ionizing Radiation Technology for Biomedical Applications" is an exciting concept. The use of hydrogels in biomedical research has been well known for a couple of decades yet the use of ionizing radiation-assisted hydrogel preparation is a novel method that needed to be promoted. Despite the authors choosing a sound topic, the manuscript is poorly written. The manuscript in the present form does not comply with the polymer journal quality. The authors are requested to consult a native language speaker to get a grammatical correction and technical rewrites and then resubmit to be considered.  

Author Response

Dear reviewer, thank you very much for your valuable comments on this review. Here are the changes I made. Point 1: The review article titled "Recent Advances on Smart Hydrogels Prepared by Ionizing Radiation Technology for Biomedical Applications" is an exciting concept. The use of hydrogels in biomedical research has been well known for a couple of decades yet the use of ionizing radiation-assisted hydrogel preparation is a novel method that needed to be promoted. Despite the authors choosing a sound topic, the manuscript is poorly written. The manuscript in the present form does not comply with the polymer journal quality. The authors are requested to consult a native language speaker to get a grammatical correction and technical rewrites and then resubmit to be considered. Response 1: We have revised the English version of the full text according to your requirements.

Round 2

Reviewer 2 Report

Recent Advances on Smart Hydrogels Prepared by Ionizing Radiation Technology for Biomedical Applications by Yang et al is a well written paper with good figures.  Here are some suggestions from my end

Section 4 : If possible add a summary table summarizing different smart hydrogels , their preparation, working and application.

Figure 5 : kindly elaborate on the caption. It is too short

Figure 3B : figure is not very clear or explanatory. You can try to change it

Kindly add more recent references in application section. Many references are dated please add recent ones.

If possible please add few future perspectives as well.

Author Response

Dear reviewer, thank you very much for your valuable comments on this review. Here are the changes I made. Point 1: Section 4 : If possible add a summary table summarizing different smart hydrogels , their preparation, working and application. Response 1: According to your request, we have added a table in the fourth part (Table 3). Point 2: Figure 5 : kindly elaborate on the caption. It is too short. Response 2: We have described it in detail. Let's change it to "Figure 5. Schematic illustration of reversible bending of an iron gel: (a) and (b) show the bending degree of field off and on respectively [52]". Point 3: Figure 3B : figure is not very clear or explanatory. You can try to change it Response 3: Thank you very much for your opinion. We have replaced Figure 3B with a study picture of drug sustained release, and made corresponding modifications in the article. Point 4: Kindly add more recent references in application section. Many references are dated please add recent ones. Response 4: Thank you very much for your advice. However, because ionizing radiation equipment is expensive and not popular, there are few literatures on biointelligent hydrogels. I am very sorry that no recent articles in this review can be found after searching. Point 5: If possible please add few future perspectives as well. Response 5: As per your request, we have added the outlook for biosmart hydrogels to the conclusion section.

Reviewer 3 Report

The authors have put some effort into editing the manuscript. However, it failed to address the majority of the concerns. The language and expressions are unclear and require technical editing and rewrites. Regret to let you know that, the manuscript in this present form is not recommended for publishing. 

Author Response

Dear reviewer, thank you very much for your valuable comments on this review. Here are the changes I made. Point 1: The authors have put some effort into editing the manuscript. However, it failed to address the majority of the concerns. The language and expressions are unclear and require technical editing and rewrites. Regret to let you know that, the manuscript in this present form is not recommended for publishing.  Response 1: Thank you very much for your advice. We have tried our best to revise the article.

Round 3

Reviewer 2 Report

The authors have addressed queries raised by the reviewers. 

Reviewer 3 Report

The response by Yang and co-workers still fails to recognize and address the flaws in the manuscript.

For E.g.  1) Introduction line 2 the author talks about horses' temples to stitch wounds. This term does not seem to be derived from standard language used in scientific conversations. The authors have used a similar style in describing certain

2) Radiation cross-linking has the advantages of simple operation, room temperature reaction, high efficiency, and greenness: I presume the author refers to green synthesis but mentioning it as greenness would not be perceived as green synthesis by the majority of the readers.

3) Page 6 last paragraph: Gel particles containing the drugs were released in a solution of pH = 1 or 7.4, which could be used to treat local diseases. The term local disease does imply anything meaningful.

 4) The pharmaceutical industry is the lifeblood of the country. This statement is too general.

5) The author should follow the proper use of et al in a manuscript. Chen [22] et al., instead mention Chen et al. and include the reference [22] at the end of the sentence.

To summarize, I regret to say that the overall quality of the manuscript is inadequate to be approved and published on Polymers. I suggest rejecting its publication and encouraging the authors to submit a revised version separately.

Author Response

Dear reviewer, thank you very much for your comments on this review. We have corrected the points you raised one by one. Details are below. Point 1: Introduction line 2 the author talks about horses' temples to stitch wounds. This term does not seem to be derived from standard language used in scientific conversations. The authors have used a similar style in describing certain. Response 1: I agree with your method very much and have revised the first paragraph of the article. Point 2: Radiation cross-linking has the advantages of simple operation, room temperature reaction, high efficiency, and greenness: I presume the author refers to green synthesis but mentioning it as greenness would not be perceived as green synthesis by the majority of the readers. Response 2: I agree with you and have deleted the word from the text. Point 3: Page 6 last paragraph: Gel particles containing the drugs were released in a solution of pH = 1 or 7.4, which could be used to treat local diseases. The term local disease does imply anything meaningful. Response 3: Thank you very much for pointing out the question, I have changed the text to read: which could be used to topical treatment Point 4: The pharmaceutical industry is the lifeblood of the country. This statement is too general. Response 4: I agree with your suggestion. I have replaced the words in the text with: Medical device is one of the core components of medical engineering. Point 5: The author should follow the proper use of et al in a manuscript. Chen [22] et al., instead mention Chen et al. And include the reference [22] at the end of the sentence. Response 5: I thank you for pointing out the mistake. I have corrected this error throughout the text.
